# Enantioselective preparation of mechanically planar chiral rotaxanes by kinetic resolution strategy

Ayumi Imayoshi [1,3], Bhatraju Vasantha Lakshmi [1], Yoshihiro Ueda [1], Tomoyuki Yoshimura [2], Aki Matayoshi [1], Takumi Furuta [1,4] & Takeo Kawabata [1✉]

Asymmetric synthesis of mechanically planar chiral rotaxanes and topologically chiral catenanes has been a long-standing challenge in organic synthesis. Recently, an excellent strategy was developed based on diastereomeric synthesis of rotaxanes and catenanes with mechanical chirality followed by removal of the chiral auxiliary. On the other hand, its enantioselective approach has been quite limited. Here, we report enantioselective preparation of mechanically planar chiral rotaxanes by kinetic resolution of the racemates via remote asymmetric acylation of a hydroxy group in the axis component, which provides an unreacted enantiomer in up to >99.9% ee in 29% yield (the theoretical maximum yield of kinetic resolution of racemate is 50%). While the rotaxane molecules are expected to have conformational complexity, our original catalysts enabled to discriminate the mechanical chirality of the rotaxanes efficiently with the selectivity factors in up to 16.

[1] Institute for Chemical Research, Kyoto University, Uji, Kyoto 611-0011, Japan. [2] Faculty of Pharmaceutical Sciences, Institute of Medical, Pharmaceutical, and Health Sciences, Kanazawa University, Kakuma-machi, Kanazawa 920-1192, Japan. [3] Present address: Graduate School of Life and Environmental Sciences, Kyoto Prefectural University, Shimogamo, Sakyo-ku, Kyoto 606-8522, Japan. [4] Present address: Department of Pharmaceutical Chemistry, Kyoto Pharmaceutical University, 1 Misasagi-Shichonocho, Yamashina, Kyoto 607-8412, Japan. ✉email: kawabata@scl.kyoto-u.ac.jp

Asymmetric synthesis of molecules with central, axial, planar, and helical chirality has been extensively developed, and the field of asymmetric synthesis is becoming a mature science. On the other hand, asymmetric synthesis of chiral mechanically interlocked molecules such as mechanically planar chiral rotaxanes and topologically chiral catenanes (Fig. 1a) has been rather unexplored. It has been proposed that rotaxanes and catenanes may possess mechanical chirality[1,2]. Even if the ring component and the axis component are not chiral by themselves, the formation of interlocked molecules consisting of ring and/or axis components brings mechanical chirality when each of the components exhibits structural dissymmetry (Fig. 1a). This was eventually demonstrated by Okamoto, Sauvage, and co-workers[3], and also by Okamoto, Vögtle, and co-workers[4] based on the circular dichroism (CD) spectra of both enantiomers of topologically chiral catenanes and mechanically planar chiral rotaxanes obtained by high-performance liquid chromatography (HPLC) separation of the racemates with chiral stationary phases.

Mechanically interlocked molecules such as rotaxanes and catenanes have attracted considerable attentions because of the non-conventional molecular frameworks[5]. Their synthesis and the applications to molecular devices and catalysis have been actively

studied[6–14]. Since the proposal of mechanically planar chirality and topological chirality in the structure of rotaxanes and catenanes, respectively[1,2], their enantioselective preparation has been a challenge in organic synthesis[14]. In most cases, the strategy largely relied on separation of enantiomers by HPLC with chiral stationary phases[3,4,6]. Recently, Goldup and co-workers proposed an excellent practical method for the preparation of an enantiomeric pair of mechanically planar chiral rotaxanes via separation of the corresponding diastereomers followed by removal of the chiral auxiliary[14–16]. The strategy was also applied to the stereoselective synthesis of topologically chiral catenanes[17]. On the other hand, a pioneering approach toward the direct catalytic enantioselective synthesis of a mechanically planar chiral rotaxane was reported by Takata and co-workers (Fig. 1b)[18]. An acylative end-capping of a pre-rotaxane complex via dynamic kinetic resolution was demonstrated to constitute a promising approach toward this goal, albeit in a low enantioselectivity (4.4% ee). Very recently, Leigh and co-workers achieved enantioselective synthesis of mechanically planar chiral rotaxanes in up to 50% ee by a chiral leaving group strategy[19]. To date, however, there have been no reports on highly enantioselective catalytic synthesis of chiral mechanically interlocked molecules. Here, we report highly enantioselective preparation of

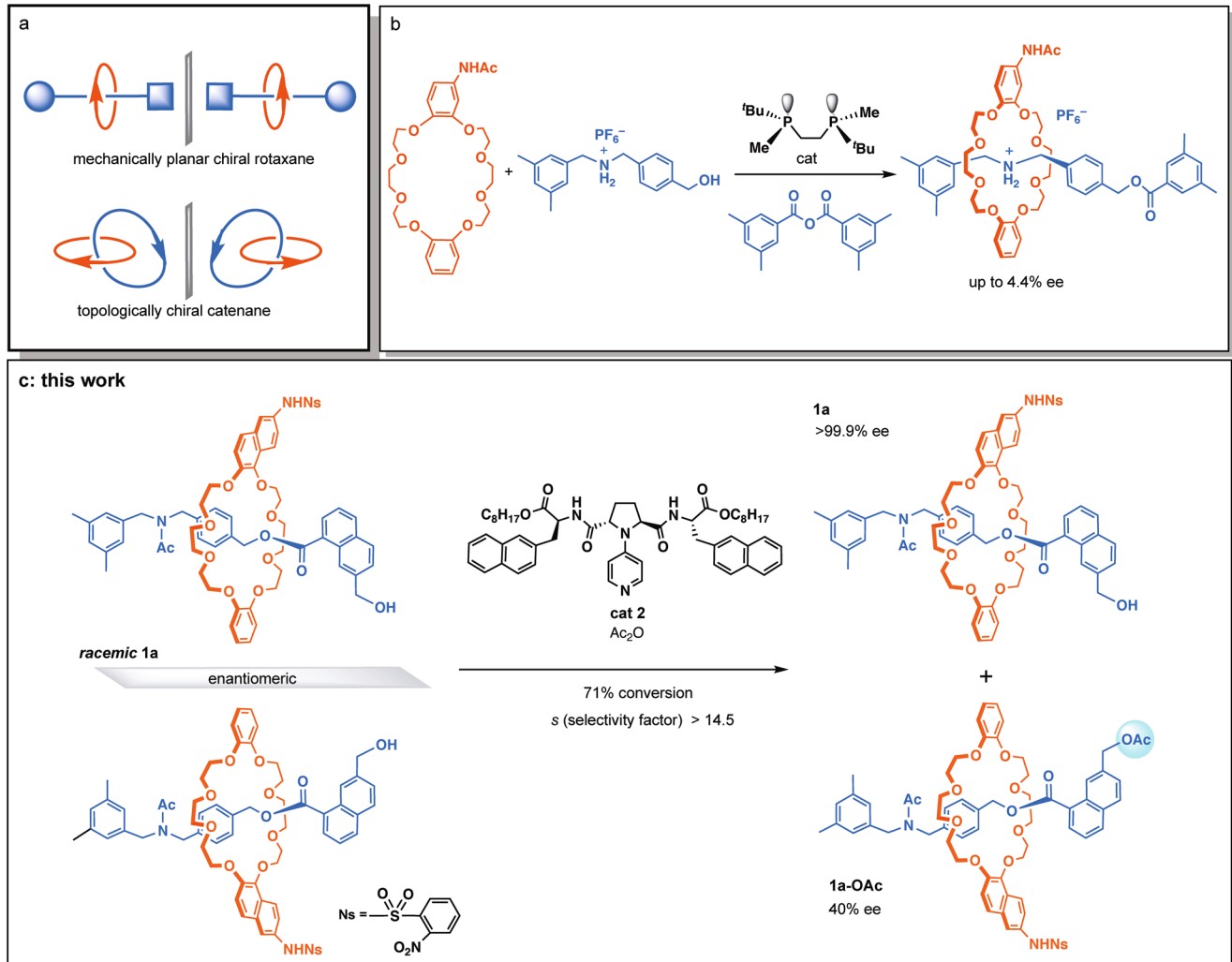

**Fig. 1 Chirality in mechanically interlocked molecules and its construction. a** A schematic presentation of a mechanically planar chiral rotaxane and a topologically chiral catenane. **b** Takata's pioneering approach toward catalytic enantioselective synthesis of a mechanically planar chiral rotaxane. **c** This work: Highly efficient acylative kinetic resolution of racemic mechanically planar chiral rotaxane **1a**. The absolute configurations of the acylated product and the recovered rotaxane alcohol were not determined. The ring component and the axis component of rotaxanes are shown in red and blue, respectively, throughout the text.

mechanically planar chiral rotaxanes by kinetic resolution of the racemates promoted by acylation catalysts (Fig. 1c). A rotaxane with >99.9% ee was successfully obtained in 29% yield at 71% conversion (selectivity factor: $s > 14.5$: $s = k_{fast}/k_{slow}$ (reaction rate of the fast-reacting enantiomer/reaction rate of the slow-reacting enantiomer)).

## Results

**Strategy for kinetic resolution.** Inspired by Takata's pioneering work in this field (Fig. 1b), we designed our strategy for enantioselective construction of mechanically planar chiral rotaxanes based on kinetic resolution of the racemate. A schematic illustration of the strategy is presented in Fig. 2a. We envisaged that asymmetric acylation of the hydroxy group of the racemic rotaxanes in the presence of a chiral catalyst would yield the acylated rotaxane and the recovered rotaxane alcohol in an enantioenriched form. The expected merits and problems with this strategy are as follows. The primary merit is that, as a general consequence of kinetic resolution, almost enantiopure recovered materials are expected to be obtained in compensation for the chemical yield[20]. A problem is determining how can such remote asymmetric induction be achieved? Methods for acylative kinetic resolution of racemic alcohols have been extensively investigated, and efficient discrimination of chirality of substrate alcohols has been achieved when the reacting center (OH) in the substrate is close to the chiral center (Fig. 2b)[21]. On the other hand, discrimination of the chirality of the substrate by acylation of the hydroxy group located at the edge of the huge mechanical interlocked molecule may not be readily achievable (Fig. 2a). Furthermore, the difficulties in discrimination of mechanical planar chirality of rotaxanes associated with the conformational flexibility and inhomogeneity may be expected (See SI for spectra indicating conformational complexity of rotaxanes: [13]C NMR of rotaxanes (conformational mixture) (S25-34), temperature-dependent [1]H NMR spectra (S58-60) and temperature-dependent CD spectra (S7) of 1a-OAc). On the other hand, a highly successful example for remote asymmetric acylation of a bisphenol compound with rigid conformation has been reported[22]. Dynamic kinetic resolution of a [2]rotaxane with point prochirality by asymmetric acylation has also been reported[23].

Aiming at effective kinetic resolution, we designed rotaxane **1a–c** (Fig. 2c) that consists of a structurally dissymmetric axis component with a hydroxy group and a structurally dissymmetric ring component with an NHNs (Ns = -SO$_2$-2-NO$_2$-C$_6$H$_4$) group. According to our previous studies[24–27], we hypothesized that use of a substrate alcohol with an NHNs group in the presence of catalyst **3** (Fig. 2d) and the analogous catalysts are the key to achieve remote asymmetric acylation. For example, we previously reported reversal of chemoselectivity in the acylation of diols **4** (Fig. 2d)[24]. Highly chemoselective acylation of the secondary hydroxy group of (5S)-**4** took place in the presence of catalyst **3**,

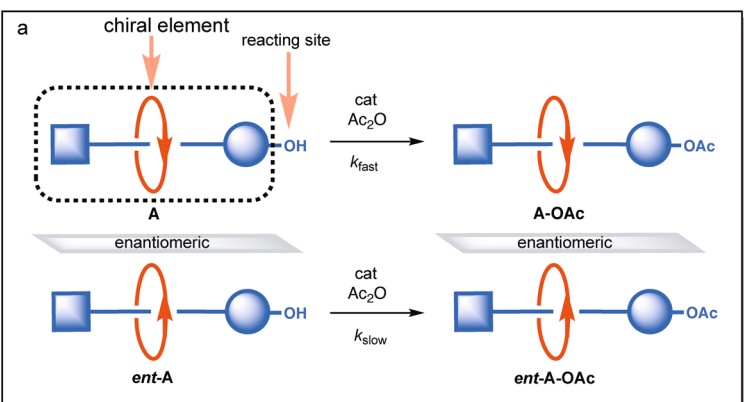

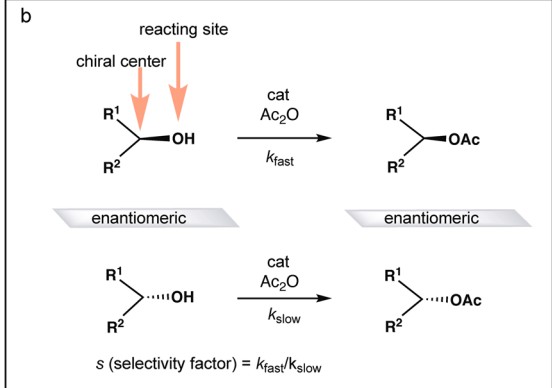

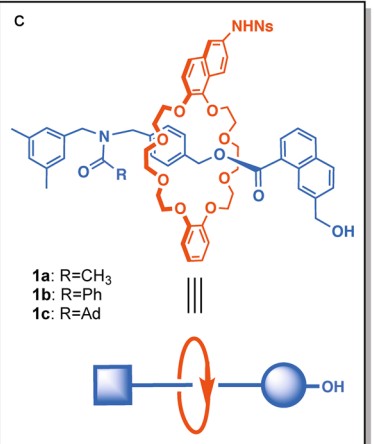

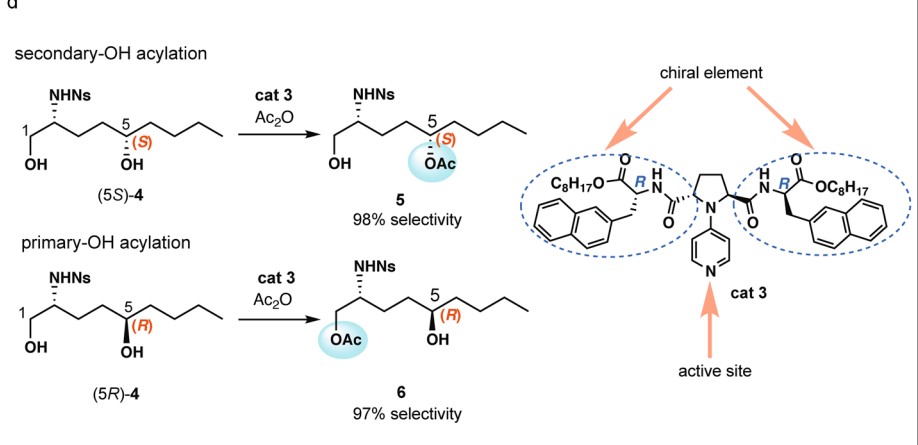

**Fig. 2 Strategy for kinetic resolution of racemic rotaxanes. a** Uncommon acylative kinetic resolution based on the recognition of remote chirality: Chirality of the substrate is hardly discriminated by the catalyst and/or the reagent upon asymmetric acylation when the chiral element is distal from the reacting center (OH). **b** Common acylative kinetic resolution based on the recognition of proximal chirality: The substrate chirality is effectively discriminated by the catalyst and/or the reagent upon asymmetric acylation when the chiral center is located close to the reacting center (OH). **c** Structure of target mechanically planar chiral rotaxane and its schematic presentation. **d** A hint for remote asymmetric acylation from our previous work: Contrasting chemoselectivity in the acylation of (5S)-**4** and (5R)-**4** with catalyst **3**.

while acylation of (5R)-**4** occurred at the primary hydroxy group almost exclusively. Comparison of the contrasting chemoselectivities suggests that catalyst **3** might recognize remote chirality at C(5) of **4**, even when acylation takes place at the hydroxy group at C(1). Catalyst **3** possesses uncommon structural characteristics as chiral catalysts, where chiral elements are located at a certain distance from the catalytically active site (pyridine nitrogen)[28]. This could be somewhat suitable for remote asymmetric acylation as shown in a possible transition state model (*vide infra*: Fig. 4b). Based on these backgrounds and other results regarding chemoselective acylation observed in our laboratory[24–26], we chose racemic rotaxanes **1a–c** and catalyst **3** and the related catalysts for the study of the kinetic resolution.

**Optimization for the conditions for kinetic resolution.** Kinetic resolution of racemic rotaxanes **1a–c** with various chiral 4-pyrroridinopyridine (PPY) catalysts **2** (Fig. 1c), **3**, **7**, and **8**[29] (Fig. 3) by asymmetric acylation was examined (Table 1). Substrates **1a–c** (Fig. 2c) were prepared according to Takata's procedure for rotaxane synthesis[30]. *Racemic* **1a** was treated with 0.9 equivalents of acetic anhydride in the presence of 0.5 equivalents of the catalyst and 1.7 equivalents of collidine as an auxiliary base in CHCl$_3$ at −60 °C at a substrate concentration of 0.07 M (Table 1, entries 1–4). Among the catalysts examined, catalyst **2** gave the best result. Kinetic resolution of *racemic* **1a** proceeded to give recovered **1a** in 77% ee at 83% conversion, which corresponds to a selectivity factor (s) of 2.6 (entry 1). With these promising results in hand, we next examined the substrate structure. The bulkiness of the amide groups (R group in **1** in Fig. 2c) in the axis component was expected to affect the relative position of the axis to the ring component according to related studies on rotaxanes[30]. While substrates with bulky amide groups such as a phenyl amide **1b** and a 1-adamantyl amide **1c** were examined, the bulkiness of the amide groups did not provide any positive effects on the efficiency of the kinetic resolution (s = 1.7 for **1b** (R = Ph), entry 5: s = 1.9 for **1c** (R = Ad), entry 6: s = 2.6 for **1a** (R = Me), entry 1). After finding that catalyst **2** is the most effective for the acylative kinetic resolution of mechanically planar chiral rotaxanes, we re-investigated the reaction parameters, including the solvents, acylating agents, auxiliary bases and concentrations (Supplementary Table 1). We employed acid anhydrides as acyl donors throughout the screening because carboxylate anion generated from the anhydride and the nucleophilic catalyst was expected to be critically important for both selectivity

and acceleration of the site-selective acylation[26,27,31,32]. It was found that selectivity of the acylation was strongly dependent on the substrate concentration. Since the best results were obtained with substrate **1a** at 0.01 M in the presence of collidine and acetic anhydride in CHCl$_3$ solution, further optimization of the reaction conditions was performed with the substrate concentration of 0.01 M (Table 2). Improved results were obtained by treatment of *racemic* **1a** (0.01 M) with 0.5 equivalents of acetic anhydride in the presence of 0.5 equivalents of catalyst **2** and 1.2 equivalents of collidine in CHCl$_3$ at −60 °C for 6 h to give the acylated chiral rotaxane in 66% ee and recovered chiral rotaxane **1a** in 23% ee, which corresponds to s = 6.1 at 26% conversion (Table 2, entry 4 vs. Table 1, entry 1). It has been often observed that lower substrate conditions like 0.01 M gave better selectivity in chemoselective acylation of diols[24] and geometry-selective acylation of tetra-substituted α,α-alkenediols[25]. We assume that substrate monomers existing in a larger ratio at the lower concentration might be more favorable for (multi-)hydrogen bonding interaction with a catalyst for the molecular recognition process (For a possible interaction, see Fig. 4b) than substrate dimers (hydrogen-bonded aggregate) existing in a larger ratio at the higher concentration.

In the course of optimizing the conditions for the kinetic resolution, we found that enantioselectivity of the kinetic resolution depended on catalyst loading. The reactions with 0.05 and 0.2 equivalents of catalyst **2** resulted in s factors of 2.2 and 4.2, respectively (Table 2, entries 2 and 3). Use of larger amounts of the catalyst resulted in an increase in asymmetric induction. Treatment of *racemic* **1a** with 0.5 equivalents of acetic anhydride in the presence of 1.5 equivalents of catalyst **2** gave the acylated chiral rotaxane in 79% ee and recovered chiral rotaxane **1a** in 63% ee, which corresponds to s = 16.1 at 44% conversion (Table 2, entry 5). Further increase in the catalyst loading to 2.0 equivalents resulted in a decrease in the efficiency of the kinetic resolution (s = 7.8, Table 2, entry 7). The optimized protocol for the kinetic resolution of **1a** was applied to the preparation of enantiopure mechanically planar chiral rotaxane. Treatment of 50 mg of **1a** with 1.5 equivalents of catalyst **2** and 0.8 equivalents of acetic anhydride in the presence of 1.2 equivalents of collidine in CHCl$_3$ at −60 °C for 37 h gave 15 mg of enantiopure rotaxane (>99.9% ee) with an s factor of >14.5 (Table 2, entry 6 and Fig. 1c: See methods and S3 for details). The efficiency of the present method for the kinetic resolution of racemic mechanically planar chiral rotaxanes is apparent when it is compared to the results using Birman's catalyst **9** (Table 2, entry 8). Birman's catalyst **9** is

**Table 1 Kinetic resolution of racemic rotaxanes: effects of the substrate structure and catalysts on the selectivity of kinetic resolution.**

<div align="center">

catalyst
Ac$_2$O
2,4,6-collidine

*racemic* 1  $\xrightarrow{\text{CHCl}_3, -60\ °C}$  enantioenriched 1  +  enantioenriched 1-OAc

</div>

| Entry | Catalyst | Substrate | Concentration (M) | Catalyst loading (equiv) | Ac$_2$O (equiv) | Collidine (equiv) | Time (h) | Ee of recovered 1 (%)[c] | Ee of 1-OAc (%)[c] | Conversion[a] (%) | s[b] |
|---|---|---|---|---|---|---|---|---|---|---|---|
| 1 | **2** | **1a** (R=Me) | 0.07 | 0.5 | 0.9 | 1.7 | 37 | 77 | 15 | 83 | 2.6 |
| 2 | **3** | **1a** (R=Me) | 0.07 | 0.5 | 0.9 | 1.7 | 34 | 19 | 9 | 67 | 1.4 |
| 3 | **7** | **1a** (R=Me) | 0.07 | 0.5 | 0.9 | 1.7 | 37 | 22 | 4 | 84 | 1.3 |
| 4 | **8** | **1a** (R=Me) | 0.07 | 0.5 | 0.9 | 1.7 | 37 | −39 | −7 | 85 | 1.5 |
| 5 | **2** | **1b** (R=Ph) | 0.07 | 0.5 | 0.9 | 1.7 | 46 | 43 | 10 | 81 | 1.7 |
| 6 | **2** | **1c** (R=Ad) | 0.07 | 0.5 | 0.9 | 1.7 | 41 | 65 | 8 | 89 | 1.9 |

[a]Conversion was determined from ee values of **1** and **1-OAc**.
[b]S-value was determined from ee values of **1** and **1-OAc**.
[c]The absolute configuration of recovered **1** and **1-OAc** was not determined.

**Table 2 Kinetic resolution of racemic rotaxane 1a: effects of catalyst loading and Birman's catalyst on the selectivity of kinetic resolution of 1a at 0.01 M substrate concentration.**

$$\text{racemic } \mathbf{1} \xrightarrow[\text{CHCl}_3, -60\,°C]{\substack{\text{catalyst} \\ \text{Ac}_2\text{O} \\ \text{2,4,6-collidine}}} \text{enantioenriched } \mathbf{1} \quad + \quad \text{enantioenriched } \mathbf{1\text{-}OAc}$$

| Entry | Catalyst | Substrate | Concentration (M) | Catalyst Loading (equiv) | Ac$_2$O (equiv) | Collidine (equiv) | Time (h) | Ee of Recovered 1 (%)[c] | Ee of 1-OAc (%)[c] | Conversion[a] (%) | s[b] |
|---|---|---|---|---|---|---|---|---|---|---|---|
| 1 | 2 | 1a (R=Me) | 0.01 | 0 | 0.5 | 1.2 | 6 | – | – | <1 | – |
| 2 | 2 | 1a (R=Me) | 0.01 | 0.05 | 0.5 | 1.2 | 6 | 1 | 38 | 3 | 2.2 |
| 3 | 2 | 1a (R=Me) | 0.01 | 0.2 | 0.5 | 1.2 | 6 | 5 | 60 | 9 | 4.2 |
| 4 | 2 | 1a (R=Me) | 0.01 | 0.5 | 0.5 | 1.2 | 6 | 23 | 66 | 26 | 6.1 |
| 5 | 2 | 1a (R=Me) | 0.01 | 1.5 | 0.5 | 1.2 | 6 | 63 | 79 | 44 | 16.1 |
| 6 | 2 | 1a (R=Me) | 0.01 | 1.5 | 0.8 | 1.2 | 37 | >99.9 | 40 | 71 | >14.5 |
| 7 | 2 | 1a (R=Me) | 0.01 | 2.0 | 0.5 | 1.2 | 6 | 58 | 63 | 48 | 7.8 |
| 8 | 9 | 1a (R=Me) | 0.01 | 1.5 | 0.5 | 1.2 | 37 | ~1 | ~1 | 29[d] | 1.0 |

[a]Conversion was determined from ee values of **1** and **1-OAc**.
[b]S-value was determined from ee values of **1** and **1-OAc**.
[c]The absolute configuration of recovered **1** and **1-OAc** was not determined.
[d]Conversion was determined from the ratio of **1** to **1-OAc**.

**Fig. 3 Structure of catalysts.** Catalysts **7** and **8** are the analogues of catalysts **2** and **3** with different side chains for substrate recognition. Birman's catalyst **9** is a representative catalyst for acylative kinetic resolution of racemic alcohols.

considered to be a highly competent catalyst for the acylative kinetic resolution of racemic alcohols, with reported s factors as high as 355[33]. However, catalyst **9** was ineffective for the kinetic resolution of **1a**, and gave recovered **1a** and the acylated **1a-OAc**, both as racemates with an s factor of 1.0 (Table 2, entry 8).

## Discussion

The s factors of the kinetic resolution of **1a** with catalyst **2** was found to depend on the catalyst loading (Table 2). Generally, the s factor of kinetic resolution is constant and independent from catalyst loading, when the contribution of the uncatalyzed process is negligible[20]. Owing to the lack of an uncatalyzed process under the present conditions for the kinetic resolution (Table 2, entry 1), the observation constitutes an unusual phenomenon. This could be resulting from mechanistic complexity, although the mechanistic details are totally unclear.

To gain mechanistic insights, the effects of the NHNs group of **1a** on the efficiency of kinetic resolution were examined (Fig. 4a). Kinetic resolution of racemic rotaxane **10**, possessing an NMeNs group instead of the NHNs group of **1a**, was performed under the optimum conditions for **1a** (Table 2, entry 5) except for the substrate, resulting in much reduced selectivity (s = 1.3) compared to the case of **1a** (s = 16.1). The reaction of **10** was also found to be quite sluggish compared with that of **1a**. Acylation of **1a** proceeded at 44% conversion after 6 h, whereas that of **10** progressed only in 8% conversion even after 37 h under the identical conditions. This suggests that the NHNs group of **1a** plays a key role in the molecular recognition process between the catalyst and the substrate as a hydrogen-bond donor. The hydrogen bonding interaction is assumed to be responsible for asymmetric induction, as well as acceleration of the acylation reaction (See Fig. 4b for a possible

transition state model, **A**). A diminished enantioselectivity (s = 3.6) was also observed in the kinetic resolution of **11** with CH$_2$NHNs group instead of an NHNs group of **1a**. This may suggest either that a more acidic hydrogen of the NHNs group in **1a** than that in **11** serves as a better hydrogen-bond donor and/or importance of the distance between the OH and NH groups for effective chiral discrimination by the catalyst. The enhanced conformational flexibility by one-carbon elongation could also be responsible for the diminished selectivity. Based on these results and previously proposed transition state models for site-selective acylation with catalyst **2** and the related catalysts[24–27,32,34], we propose **A** as a possible transition state model for remote asymmetric acylation (Fig. 4b). Structure **A** was generated by an ONIOM method (M06-2X/6-31 + G**:PM6: For details, see S12) with constraint of hydrogen bonds indicated by yellow circles (Fig. 4b, right illustration) on the assumption that hydrogen bonding interaction between the amide carbonyl group of catalyst **2** and NsN-H[24–26], as well as general base catalysis by the carboxylate anion located in proximity to the reacting hydroxy group[32] are the keys for the effective remote asymmetric acylation. Note that transition state model **A** is generated based on the above assumption, and it does not mean the most stable transition state structure for the acylation of **1a**.

In conclusion, a mechanically planar chiral rotaxane with >99.9% ee has been prepared by kinetic resolution of the racemate via asymmetric acylation. Efficient discrimination of mechanical planar chirality of the rotaxane was achieved by unusually effective remote asymmetric acylation of the hydroxy group located at the edge of the rotaxane molecule via precise molecular recognition with the catalyst. This phenomenon should provide unique aspects in asymmetric synthesis and molecular recognition. It would accelerate potential applications of mechanical chirality in catalysis[6–13]. Recently, application of rotaxanes to medicinal chemistry has been

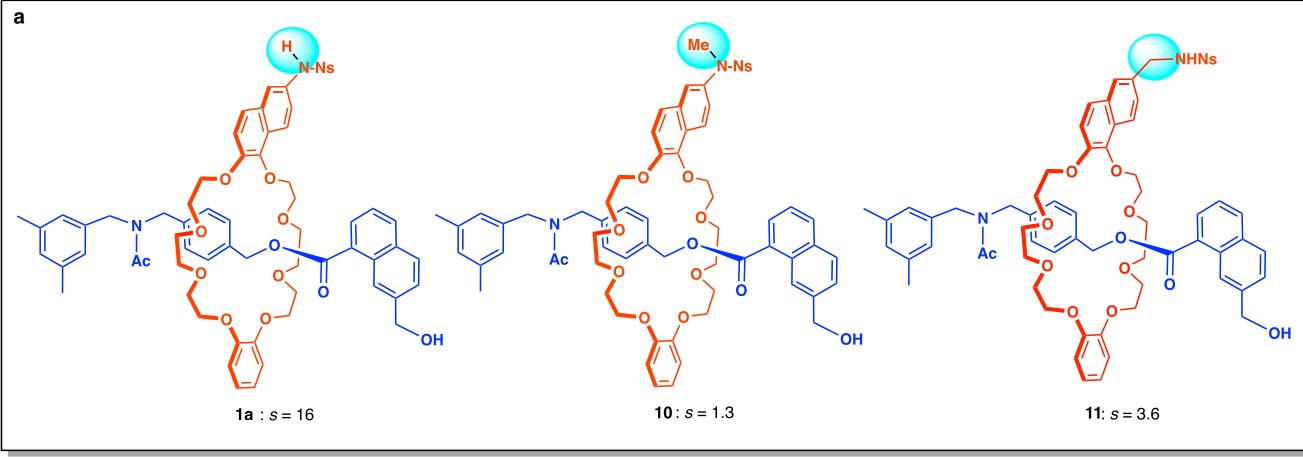

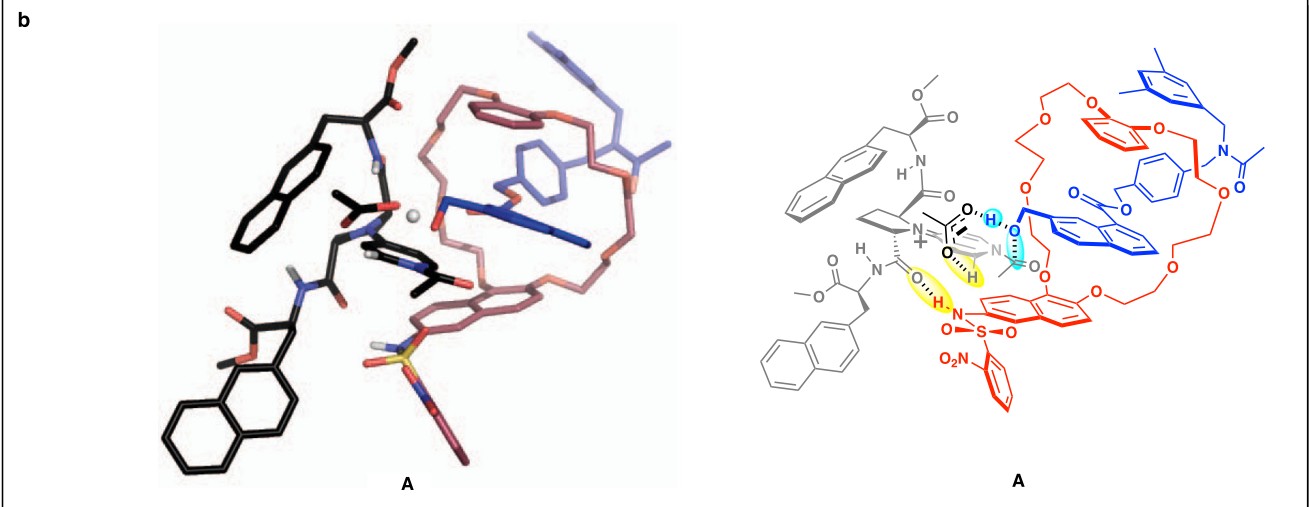

**Fig. 4 Mechanistic implication for remote asymmetric acylation. a** Relationship between substrate structure and the selectivity factor (*s*) of the kinetic resolution. The absolute configuration of recovered rotaxanes was not determined. **b** left: A calculated transition state model **A** for remote asymmetric acylation of rotaxane alcohol **1a** with a catalyst model of **2** (the corresponding methyl ester of **2**). Right: a schematic drawing of model A in which catalyst, ring component, axis component, and carboxylate anion are shown in grey, red, blue, and black, respectively. Key hydrogen bonds are indicated by yellow circles. Structure **A** was generated by an ONIOM method (see S12) with constraint of hydrogen bonds indicated by yellow circles. Note that transition state model **A** is generated based on the assumption above, and it does not mean the most stable transition state structure.

emerging. Rotaxanes have been shown to have biologically desirable properties[35–37]. However, to the best of our knowledge, enantiopure rotaxanes with mechanical planar chirality have never been examined in the studies directed toward medicinal chemistry. As biological activity is to be dependent on the enantiomeric structure of the molecule, development of the methods for the preparation of mechanically planar chiral rotaxanes with high-enantiomeric purity is expected to contribute to create an additional chemical space in medicinal chemistry.

## Methods

**Procedures for preparation of enantiopure mechanically planar chiral rotaxane 1a.** To a CHCl₃ solution (2.25 ml) of catalyst **2** (54.3 mg, 0.0635 mmol, 1.5 equivalents) were added a CHCl₃ solution of 2,4,6-collidine (0.075 M, 0.67 ml, 0.0503 mmol, 1.2 equivalents) and a CHCl₃ solution (1.0 ml) of *racemic* rotaxane **1a** (50.0 mg, 0.0424 mmol, 1.0 equivalents) at room temperature. The mixture was cooled to −60 °C. A CHCl₃ solution of acetic anhydride (0.105 M, 0.32 ml, 0.0336 mmol, 0.8 equivalents) was added dropwise to the mixture, and the reaction mixture was stirred at −60 °C for 37 h. Then MeOH (15 ml) was added to the reaction mixture at −60 °C, and the mixture was stirred at room temperature for 30 min. The mixture was evaporated to dryness under a reduced pressure. The crude product was subjected to preparative thin layer chromatography (TLC) (SiO₂, MeOH/CHCl₃ 1/19) to allow isolation of recovered **1a** (15.0 mg, 29% yield) and acylated **1a-OAc** (33.0 mg, 65% yield). Enantiomeric purity of the recovered

rotaxane, **1a** (>99.9% ee), and the acylated product, **1a-OAc** (40% ee), was determined by chiral HPLC with CHIRALPAK-IC column eluted by EtOH/CH₂Cl₂ 1/60 (0.7 ml/min) at 20 °C. Conversion (C, C = 71%) and selectivity factor (*s*, *s* > 14.5) were determined according to the following equation: conversion $C = ee_{1a}/(ee_{1a} + ee_{1a\text{-}OAc})$ and selectivity factor $s = \ln[(1 − C)(1 − ee_{1a})]/\ln[(1 − C) (1 + ee_{1a})] = \ln [1 − C(1 + ee_{1a\text{-}OAc})]/\ln[1 − C(1 − ee_{1a\text{-}OAc})]$.

**Reporting summary**. Further information on research design is available in the Nature Research Reporting Summary linked to this article.

## Data availability

The data supporting the findings of this study are available within the article and its Supplementary Information File. Any further relevant data are available from the authors on request.

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

## Acknowledgements

This research was financially supported by Grants-in-Aids for Scientific Research (S) (JP26221301), Young Scientists (B) (JP15K18827), and Scientific Research on Innovative Areas "Advanced Molecular Transformations by Organocatalysts" (JP23105008) and "Middle Molecule Strategy" (JP16H01148). A.I. acknowledges the financial support through JSPS Research Fellowships for Young Scientists (JP15J10954).

## Author contributions

T.K. conceived the work; A.I., B.V.L., and T.K. devised the experiments; A.I. carried out the major part of the experiments; B.V.L., Y.U., and A.M. carried out the experiments; Y.U., T.Y., and T.F. supported analyses of data; Y.U. performed the computational study; A.I. and T.K. wrote the paper.

## Competing interests

The authors declare no competing interests.
