## [Peer Review File · Nature Communications]

Reviewer #1 (Remarks to the Author):

In the manuscript entitled 'Enantioselective preparation of rotaxanes with inherent supramolecular chirality by kinetic resolution strategy', the authors describe the kinetic resolution of mechanically planar chiral rotaxane racemates by acylation with a chiral nucleophilic catalyst. It is possible to obtain rotaxanes with impressive ee values of >99.9%. Accessing enantioenriched samples of such rotaxanes is notoriously difficult, with most studies of these molecules relying on chiral HPLC separation or use of the racemate. Recent studies by Goldup (refs 9 and 10), Takata (ref 12) and Leigh (ref 13) have set the scene for their stereoselective synthesis.

The real challenge posed by the stereoselective synthesis of such rotaxanes is how to direct the assembly of the ring and thread components into a specific orientation, i.e. how to develop a selective rotaxane formation, rather than how to differentiate enantiomers by post-assembly modification. As this manuscript describes a post-assembly approach, it represents a complementary method to chiral HPLC separation rather than a novel strategy for rotaxane synthesis. In addition, the macrocycle screen reveals the specificity in structure required to gain high e.e. values.

The manuscript has sufficient novelty and general interest that, in my opinion, it should be published in Nature Communications. The following points should be considered prior to publication:

(1) Rotaxanes are discrete molecules, not supramolecular assemblies/supramolecules (Angew. Chem. Int. Ed. 2004, 43, 3260), so do not contain inherent supramolecular chirality. The term 'mechanical planar chirality' accurately describes the chirality dealt with here. Any reference to supramolecular chirality or supramolecules, including in the title, should be changed to reflect this.

(2) The authors state: 'The second problem is the difficulty in discrimination of supramolecular chirality associated with its mobility and diversity.' This issue is raised continually by the authors throughout the manuscript. A thorough investigation into the conformational behavior of the rotaxanes, rather than speculation, is therefore required. The authors should perform 2D NMR (NOESY, ROESY or EXSY) and VT-NMR on 1a to investigate 'conformational flexibility and diversity of mechanically planar chiral rotaxane 1a-OAc'.

(3) The manuscript has a confusing structure. I feel that the work would be more accessible to the reader if the lengthy superfluous discussion of the authors' previous work (lines 126-171) was condensed into one or two sentences. Also, the figures should be revised to convey the work more clearly than the current ones do. Schemes depicting the resolution reaction with full chemical structures and the proposed mechanism would make the work much easier to understand.

(4) Did the authors measure any binding between the catalyst and the rotaxanes? If such binding could be observed by ¹H NMR, further 2D experiments would provide more compelling evidence for the proposed mechanism.

(5) Reference 11 lacks the title of the paper.

(6) The authors should additionally cite J. Am. Chem. Soc. 2008, 130, 1836 as another example of kinetic resolution applied to rotaxanes. Two highly relevant and recent reviews — Symmetry 2020, 12, 144 and Chem. - Eur. J. 2018, 24, 3101 — should also be cited.

(7) The authors describe the chemical structures in figures as showing 'tentative absolute configuration'. As the authors cannot determine the absolute configuration of structures based on the evidence they provide, such assignments are random, rather than tentative. The description should be changed to 'unknown absolute configuration' or similar, or the authors should provide justification for the current wording.

(8) According to the equations provided, the conversion and s factor in table 1 entry 2-4 and table 2 entry 8 are calculated incorrectly. Specifically, using the equations and data provided in the manuscript, the s factor giving 99.9% e.e for the title compound (table 2, entry 6) should be 14.6, not >16!

(9) The SI needs carefully checking for errors. Many compounds do not have the correct number of carbon peaks listed in their ¹³C NMR data. For example, title compound 1a-OH has >20 more carbon peaks listed than carbon environments in the molecule. If this is due to co-conformers existing in solution, then this should be explained. This is especially important for compounds whose fluxional behaviour is discussed in detail within the manuscript.

Reviewer #2 (Remarks to the Author):

The article from Kawabata and co-workers describes a really exciting result – the first high enantioselectivity catalytic enantioselective approach to mechanically planar chiral rotaxanes. This finding is an important next step in the progress of these molecules towards increased study and applications. The chemistry itself is solid – the claims are well substantiated and documented. However, the way the manuscript is written makes it quite hard to follow what is going on. There is a very long introduction to the area, which includes a lot of preamble about the method to be used, whereas it might be better to move the results forward and shorten the introduction.

In my view this work is definitely suitable for publication in Nature Communications once the following points have been considered/addressed.

1) Introduction

The introduction is a bit rambling and hyperbolic in places (for example, it is hard to argue that "tremendous effort" has been devoted to developing methods to access chiral interlocked molecules – very few groups have published in the area, although that is changing). Some of the references are incorrect. For example, the first separation of enantiomeric topologically chiral catenanes was achieved by Sauvage and Okamoto, not Vogtle. It is also slightly odd that key reviews in the area (e.g. 23,24,26,27) are cited in the conclusions rather than in support of some of the key points in the introduction. Other reviews on chiral interlocked molecules have been published by Niemeyer, Takata and Evans. These should be cited correctly. I would also recommend that the Stoddart book on the mechanical bond is cited – it is currently the bible in the field and includes a chapter on stereochemistry. An important example of catalytic enantioselective synthesis of a co-conformationally chiral rotaxane is omitted (<https://pubs.acs.org/doi/10.1021/ja7102394>). Finally, the examples of topologically chiral catenanes and mechanically planar chiral rotaxanes containing additional covalent stereochemistry are mostly incorrect – they either don't contain a mechanical stereogenic element or are pseudorotaxanes. I think this latter point is actually just a typo – are the authors intending to highlight interlocked molecules with covalent stereochemistry?

2) Start of the R&D – overly long

The start of the R&D is not particularly concise. The key points to get across are 1) kinetic resolution of interlocked molecules could be difficult because the stereochemistry of the mechanical bond may not interact strongly with the chiral catalyst, leading to poor enantiodifferentiation, particularly if the reacting group is remote from the mechanical bond; 2) the authors have already developed some very impressive chemistry using a chiral DMAP derivative that is very good at differentiating remote stereochemistry, presumably due to specific non-covalent interactions between the catalyst and the substrate; 3) maybe, the catalyst described in (2) could solve the problem described in (1). I'm not convinced we need several diagrams and well over a page of text to get these key points across. In my opinion, these ideas could be expressed in a single paragraph in the introduction itself.

3) Start of the R&D - Conformational effects etc

The authors suggest that (co)conformational inhomogeneity (what I believe they mean by “diversity” – this word is unclear and should be changed) is a big problem in their system. This may be the case but the CD spectra etc presented don't really substantiate this. In particular, although the rotaxane shows a greater % change in molar CD than binol, the absolute change looks quite similar (although it is hard to be sure because they are plotted on very different scales – this can be solved by stating the numerical change in CD intensity). I'm generally unconvinced by these arguments, particularly as they seem to then focus on conformational motion about the benzylic C-C bond. Although this is indeed one example of conformational motion in the system, given that the catalyst interacts with the macrocycle to deliver the substrate to the axle (see below), this motion seems to be the least of the problem!

Generally I would be happier with the statement “(co)conformational effects have the potential to be very large” without all of the accompanying justification/conjecture. I'm not convinced the diagrams and discussion add anything, particularly as, beyond the effect of temperature on CD, there is no hard data behind it.

4) Origin of selectivity

I have been quite critical above but that is because I am concerned that the current presentation of the manuscript is obscuring some extremely nice results with the catalysis! The authors could lose a lot of the text at the start of the R&D and start with the results obtained and the manuscript would be much more compelling – less conjecture more hard data!

However, to contradict myself, we also need some more conjecture, specifically on the origin of the enantioselectivity. My understanding is that the authors' catalyst relies on a hydrogen bond between the NHNs group and the carbonyl of the proline-like unit on the DMAP derivative, as well as some pi-pi stacking (maybe). This isn't really made clear at all, although it is implied by some of the substrate studies and in the discussion of the authors' previous results. In a perfect world it would be nice to see some more substrate scope to try and tie this down but the world is far from perfect at the moment and I am aware these studies are very hard at the best of times. To bridge this gap, I would suggest that a very basic molecular model is provided showing what the authors think might be happening. This can be constructed with constraints etc and should be presented with clear health warnings – it is just a good illustration showing how the process *could* be working. To be clear, modelling such a complex system is a very big undertaking and I am not asking the authors to launch a major computational investigation.

To a certain extent, the proposed discussion/illustration of the origin of selectivity will solve one of the problems with the current manuscript: if the paper is supposed to be about the catalytic enantioselective synthesis of MPC rotaxanes, the lack of rotaxane substrate scope is an issue. However, if the paper is clearly about the properties of the catalyst that lead to enantioselectivity, then there needs to be a clearer explanation for the observed structure activity relationships.

5) Conclusions

The conclusions are a bit strange starting from “we anticipate...”. As mentioned before, the citations given to reviews should probably be in the introduction. The citations to molecular machines reviews lack context in the conclusions (why do we want chiral molecular machines?). The citations to applications in medicinal chemistry are odd because stereochemistry plays no role in any of the examples given (albeit the recognition of the sugar in 30 is stereoselective, but this is covalent stereochemistry). The authors also lump one of the few examples of an application of mechanical stereochemistry (26) in with a list of reviews when it is perhaps one of the few practical justifications (albeit preliminary) for studying how to make these molecules!

6) Some general points about terminology: “inherent chirality”, “supramolecular chirality” etc. The terminology in the area generally is a bit of a mess because it has grown up in a slightly ad hoc manner. I’m really not keen on “inherent chirality” as it’s a bit of a meaningless term – why is a calixarene (where the term is often used) more “inherently” chiral than an amino acid? I would also remove all mention of “supramolecular chirality” given that rotaxanes are technically not supramolecular species (covalent bonds must be broken to separate the components, thus they are not supramolecules). I have a preference for “mechanical chirality” or “co-conformational chirality” (depends on the type of stereogenic unit) to discuss the stereochemistry that is displayed by rotaxanes and catenanes as a result of the mechanical bond. On an unrelated point, I am not keen on the term “asymmetric synthesis” (I know it is widely used). A synthesis itself is not “asymmetric” (lacking in symmetry elements). It is an enantioselective synthesis.

Reviewer #3 (Remarks to the Author):

Kawabata and co-workers report upon the enantioselective preparation of mechanically planar chiral rotaxanes by use of a kinetic resolution strategy. Impressive enantiomeric excess of the unreacted enantiomer in reasonable yield has been demonstrated. A detailed supplementary information has been included containing NMR and HPLC data as evidence of work carried out.

There is considerable fundamental research interest in mechanically chiral interlocked molecules at this time - as well as the potential for exciting real-world applications as described in the conclusion.

The results here are novel and distinct from other key works on the asymmetric synthesis of mechanically chiral rotaxanes (e.g. Goldup, Leigh and Takata) and as such are of interest sufficient for publication in Nature Communications, once various points have been attended to.

Principal concerns:

The discussion of Conformers X and Y (depicted in Fig 3). How speculative is this? Do the authors need to make these claims? Would it be better to supply in Supplementary Information as a "proposal"? I would emphasise that the main conclusion of the paper regarding the successful enantioselective kinetic resolution still holds without this discussion.

I suspect the X/Y discussion has been included as the authors may have used this model to tentatively assign the absolute structures of the rotaxanes in Fig 1(?) If so this needs to be clearly stated - highlighting uncertainties. If not, the authors should still explain how they have tentatively assigned structures of enantiomers of rotaxane.

Other points to be attended to:

Line 36: To avoid confusion for the non-specialist, add the word "may" so line reads ". . . has been claimed that rotaxanes and catenanes may possess . . ."

Paragraph starting Line 53: Again, as written this paragraph could mislead the non-specialist reader. The first sentence is correct in stating that "tremendous efforts have been devoted to achieving their asymmetric synthesis". However, as written the second sentence states that construction of mechanically planar chiral rotaxanes and topologically chiral catenanes requires incorporation of additional chiral elements. This isn't true - additional chiral elements have only been required to date if attempting to prepare these molecules in mechanically chiral enriched form. This sentence (at least) needs to be re-worded.

I would also advise the authors to check that the references included (particularly Refs 5-8) are in the correct places of this revised paragraph.

Line 96 and Fig 2 - Why do the authors discuss 6 conformers (I to VI) of the rotaxane? Is there a reason for thinking there are 6?

Line 127: What is compound 2? Structure is not provided in main manuscript.

Fig 3: Correct spelling of "accessible"

Line 200: "We unexpectedly found that enantioselectivity . . . depended on catalyst loading" - is this unexpected?

Ref 11: Missing paper title.

September 18, 2020

The revised points are as follows.

For reviewer 1:

- (1) Rotaxanes are discrete molecules, not supramolecular assemblies/supramolecules (Angew. Chem. Int. Ed. 2004, 43, 3260), so do not contain inherent supramolecular chirality. The term ‘mechanical planar chirality’ accurately describes the chirality dealt with here. Any reference to supramolecular chirality or supramolecules, including in the title, should be changed to reflect this.

Response:

The term ‘inherent supramolecular chirality’ was changed to ‘mechanical chirality’, ‘mechanical planar chirality’ or “chirality”, depending on the situations.

The term ‘supramolecule’ indicating rotaxane and/or catenane was changed to ‘rotaxane’ or ‘catenane’, or ‘mechanical interlocked molecule’, depending on the situations.

- (2) The authors state: ‘The second problem is the difficulty in discrimination of supramolecular chirality associated with its mobility and diversity.’ This issue is raised continually by the authors throughout the manuscript. A thorough investigation into the conformational behavior of the rotaxanes, rather than speculation, is therefore required. The authors should perform 2D NMR (NOESY, ROESY or EXSY) and VT-NMR on 1a to investigate ‘conformational flexibility and diversity of mechanically planar chiral rotaxane 1a-OAc’.

Response:

Because the evidence of conformational flexibility and diversity of rotaxanes was not enough, we removed the detailed discussion about this issue. Instead, the issue was simply mentioned in the text as follows.

Lines 87-91:

Furthermore, the difficulties in discrimination of mechanical planar chirality of rotaxanes associated with the conformational flexibility and inhomogeneity may be expected (See

SI for spectra indicating conformational complexity of rotaxanes: ^{13}C NMR of rotaxanes (conformational mixture) (S19-28), temperature-dependent ^1H NMR spectra (S51) and temperature-dependent CD spectra (S7) of **1a-OAc**.

Original Fig. 2c with a footnote “Diversity and mobility of supramolecular chirality:” was removed.

Original Fig. 2d with a footnote “Unusually strong temperature-dependency of the CD spectra of enantiopure (+)-**1a-OAc**” was moved to SI (S7).

Temperature dependent ^1H NMR spectra of **1a-OAc** was shown in SI (S51-56) together with its HMBC and HMQC spectra.

(3) The manuscript has a confusing structure. I feel that the work would be more accessible to the reader if the lengthy superfluous discussion of the authors’ previous work (lines 126-171) was condensed into one or two sentences. Also, the figures should be revised to convey the work more clearly than the current ones do. Schemes depicting the resolution reaction with full chemical structures and the proposed mechanism would make the work much easier to understand.

Response:

(a) Original Lines 126-171 and the corresponding original Fig 3 were removed. Instead, a simplified scheme for our previous results were shown in Fig 2d. The corresponding simplified explanation of strategy for remote asymmetric acylation was shown as in lines 113-122.

We reported reversal of chemoselectivity in the acylation of diols **4** (Fig. 2d)²³. Highly chemoselective acylation of the secondary hydroxy group of (*5S*)-**4** took place in the presence of catalyst **3**, while acylation of (*5R*)-**4** occurred at the primary hydroxy group. Comparison of the contrasting chemoselectivities suggests that catalyst **3** might recognize remote chirality at C(5) of **4**, even when acylation takes place at the hydroxy group at C(1). Catalyst **3** also seems to possess promising structural characteristics for remote asymmetric acylation because the chiral elements are located far from the catalytically active site (pyridine nitrogen). Based on these backgrounds and other results regarding chemoselective acylation observed in our laboratory²⁴⁻²⁶, we chose racemic rotaxanes **1a-c** and catalyst **3** and the related catalysts for the study of the kinetic resolution.

(b) A hypothetical proposed mechanism was shown in Fig. 4b with notice “although this

is merely an illustration for one of the possible rationales” in the text (line 195).
For the consideration of the mechanism, references 29 and 30 were added.

(4) Did the authors measure any binding between the catalyst and the rotaxanes? If such binding could be observed by ^1H NMR, further 2D experiments would provide more compelling evidence for the proposed mechanism.

Response:

We tried to measure ^1H NMR spectra indicating some binding between the catalyst and the rotaxane. However, the spectra were too complicated to analyze.

(5) Reference 11 lacks the title of the paper.

Response:

The title was shown in the revised reference 16.

(6) The authors should additionally cite J. Am. Chem. Soc. 2008, 130, 1836 as another example of kinetic resolution applied to rotaxanes. Two highly relevant and recent reviews — Symmetry 2020, 12, 144 and Chem. - Eur. J. 2018, 24, 3101 — should also be cited.

Response:

These literatures were cited in references 22, 13, and 11, respectively.

(7) The authors describe the chemical structures in figures as showing ‘tentative absolute configuration’. As the authors cannot determine the absolute configuration of structures based on the evidence they provide, such assignments are random, rather than tentative. The description should be changed to ‘unknown absolute configuration’ or similar, or the authors should provide justification for the current wording.

Response:

A sentence ‘The absolute configuration was not determined’ was used in place of ‘tentative absolute configuration’.

(8) According to the equations provided, the conversion and s factor in table 1 entry 2-4 and table 2 entry 8 are calculated incorrectly. Specifically, using the equations and data provided in the manuscript, the s factor giving 99.9% e.e for the title compound (table 2, entry 6) should be 14.6, not >16!

Response:

Figure 3, Table 2, entry 6: S -factor was changed to >14.5.

The *s* factors in Table 1, entries 2-4 and Table 2, entry 8 were corrected.

The methods for determining the *s*-values and the conversion were indicated in the footnotes.

(9) The SI needs carefully checking for errors. Many compounds do not have the correct number of carbon peaks listed in their ¹³C NMR data. For example, title compound **1a-OH** has >20 more carbon peaks listed than carbon environments in the molecule. If this is due to co-conformers existing in solution, then this should be explained. This is especially important for compounds whose fluxional behavior is discussed in detail within the manuscript.

Response:

Rotaxanes **1a**, **1a-OAc**, **10**, **11**, **10-OAc**, **11-OAc**, and their synthetic precursor rotaxanes **28**, **32**, and **37** show much larger numbers of carbon peaks in ¹³C spectra than their carbon contents: **1a**: 89 peaks for C₆₅, **1a-OAc**: 76 peaks for C₆₇, **10**: 81 peaks for C₆₆, **11**: 88 peaks for C₆₆, **10-OAc**: 86 peaks for C₆₈, **11-OAc**: 93 peaks for C₆₈, **28**: 102 peaks for C₇₂, **32**: 85 peaks for C₇₁, **37**: 88 peaks for C₇₃.

This phenomena could be ascribed to the mixture of conformers. However, we cannot discriminate whether the conformers are tertiary amide *E/Z* conformers or rotaxane conformers by a shuttling process. Rotaxane conformers by the shuttling process was reported in studies of the the related rotaxane molecules (T. Takata, *et al. Chemistry Lett.* **36**, 208-209 (2007)). The issue was simply described in the text (lines 88-91).

“(See SI for spectra indicating conformational complexity of rotaxanes: ¹³C NMR of rotaxanes (conformational mixture) (S19-28), temperature-dependent ¹H NMR spectra (S51) and temperature-dependent CD spectra (S7) of **1a-OAc**).”

Thank you again for your pertinent suggestions.

Sincerely yours,

Takeo Kawabata
Institute for Chemical Research, Kyoto University
Uji, Kyoto 611-0011, Japan

September 18, 2020

For reviewer 2:

(1) Introduction

The introduction is a bit rambling and hyperbolic in places (for example, it is hard to argue that “tremendous effort” has been devoted to developing methods to access chiral interlocked molecules – very few groups have published in the area, although that is changing).

Response:

Expression was changed as shown in lines 52-53:

Since the discovery of mechanically planar chiral rotaxanes and topologically chiral catenanes, their enantioselective preparation has been a challenge in organic synthesis.

(1) Some of the references are incorrect. For example, the first separation of enantiomeric topologically chiral catenanes was achieved by Sauvage and Okamoto, not Vogtle. It is also slightly odd that key reviews in the area (e.g. 23,24,26,27) are cited in the conclusions rather than in support of some of the key points in the introduction. Other reviews on chiral interlocked molecules have been published by Niemeyer, Takata and Evans. These should be cited correctly. I would also recommend that the Stoddart book on the mechanical bond is cited – it is currently the bible in the field and includes a chapter on stereochemistry. An important example of catalytic enantioselective synthesis of a co-conformationally chiral rotaxane is omitted (<https://pubs.acs.org/doi/10.1021/ja7102394>). Finally, the examples of topologically chiral catenanes and mechanically planar chiral rotaxanes containing additional covalent stereochemistry are mostly incorrect – they either don't contain a mechanical stereogenic element or are pseudorotaxanes. I think this latter point is actually just a typo – are the authors intending to highlight interlocked molecules with covalent stereochemistry?

Response:

- (a) The first separation of enantiomeric topologically chiral catenanes reported by Sauvage and Okamoto was cited in reference 3. The original reference 4 was removed because the separated rotaxane does not possess mechanical planar chirality.
- (b) Reviews in the original references 23-27 were moved to the introductory part of the text, and cited as references 6-10.
- (c) Reviews on chiral interlocked molecules by Niemeyer, Takata and Evans were cited in references 11-13.
- (d) The Stoddart's book on the mechanical bond was cited in reference 5.
- (e) Suggested important literature (<https://pubs.acs.org/doi/10.1021/ja7102394>) was cited in reference 22.
- (f) The original references 5-8 were removed because these literatures are not proper examples for topologically chiral catenanes and mechanically planar chiral rotaxanes as the referee suggested.

(3) Start of the R&D – overly long

The start of the R&D is not particularly concise. The key points to get across are 1) kinetic resolution of interlocked molecules could be difficult because the stereochemistry of the mechanical bond may not interact strongly with the chiral catalyst, leading to poor enantiodifferentiation, particularly if the reacting group is remote from the mechanical bond; 2) the authors have already developed some very impressive chemistry using a chiral DMAP derivative that is very good at differentiating remote stereochemistry, presumably due to specific non-covalent interactions between the catalyst and the substrate; 3) maybe, the catalyst described in (2) could solve the problem described in (1). I'm not convinced we need several diagrams and well over a page of text to get these key points across. In my opinion, these ideas could be expressed in a single paragraph in the introduction itself.

Response:

The original start part of the R&D (lines 79-171) was shortened to two paragraphs and described in lines 73-94 and 108-122.

Original Fig. 3 indicating strategy for asymmetric acylation and the accompanying explanation (line 144-160) in text were removed.

(3) Start of the R&D - Conformational effects etc

The authors suggest that (co)conformational inhomogeneity (what I believe they mean by "diversity" – this word is unclear and should be changed) is a big problem in their system.

This may be the case but the CD spectra etc presented don't really substantiate this. In particular, although the rotaxane shows a greater % change in molar CD than binol, the absolute change looks quite similar (although it is hard to be sure because they are plotted on very different scales – this can be solved by stating the numerical change in CD intensity). I'm generally unconvinced by these arguments, particularly as they seem to then focus on conformational motion about the benzylic C-C bond. Although this is indeed one example of conformational motion in the system, given that the catalyst interacts with the macrocycle to deliver the substrate to the axle (see below), this motion seems to be the least of the problem!

Response:

The term “conformational diversity” was changed to “conformational inhomogeneity”. Original Fig. 2c indicating diversity and mobility of rotaxanes was removed. Temperature-dependent CD spectra (original Fig 2d) was moved to SI, and the discussion about conformational diversity in the text (lines 95-102) was removed. Instead, the issue was simply mentioned in the text as follows.

Lines 87-91:

Furthermore, the difficulties in discrimination of mechanical planar chirality of rotaxanes associated with the conformational flexibility and inhomogeneity may be expected (See SI for spectra indicating conformational complexity of rotaxanes: ^{13}C NMR of rotaxanes (conformational mixture) (S19-28), temperature-dependent ^1H NMR spectra (S51) and temperature-dependent CD spectra (S7) of **1a-OAc**).

(4) Generally I would be happier with the statement “(co)conformational effects have the potential to be very large” without all of the accompanying justification/conjecture. I'm not convinced the diagrams and discussion add anything, particularly as, beyond the effect of temperature on CD, there is no hard data behind it.

Response:

The discussion on conformational diversity was removed from the text. Instead, the following comments on conformational inhomogeneity were added in lines 87-91.

Furthermore, the difficulties in discrimination of mechanical planar chirality of rotaxanes associated with the conformational flexibility and inhomogeneity may be expected (See SI for spectra indicating conformational complexity of rotaxanes: ^{13}C NMR of rotaxanes (conformational mixture) (S19-28), temperature-dependent ^1H NMR spectra (S51) and temperature-dependent CD spectra (S7) of **1a-OAc**).

(4) Origin of selectivity

I have been quite critical above but that is because I am concerned that the current presentation of the manuscript is obscuring some extremely nice results with the catalysis! The authors could lose a lot of the text at the start of the R&D and start with the results obtained and the manuscript would be much more compelling – less conjecture more hard data!

However, to contradict myself, we also need some more conjecture, specifically on the origin of the enantioselectivity. My understanding is that the authors' catalyst relies on a hydrogen bond between the NHNs group and the carbonyl of the proline-like unit on the DMAP derivative, as well as some pi-pi stacking (maybe). This isn't really made clear at all, although it is implied by some of the substrate studies and in the discussion of the authors' previous results. In a perfect world it would be nice to see some more substrate scope to try and tie this down but the world is far from perfect at the moment and I am aware these studies are very hard at the best of times. To bridge this gap, I would suggest that a very basic molecular model is provided showing what the authors think might be happening. This can be constructed with constraints etc and should be presented with clear health warnings – it is just a good illustration showing how the process *could* be working. To be clear, modelling such a complex system is a very big undertaking and I am not asking the authors to launch a major computational investigation.

To a certain extent, the proposed discussion/illustration of the origin of selectivity will solve one of the problems with the current manuscript: if the paper is supposed to be about the catalytic enantioselective synthesis of MPC rotaxanes, the lack of rotaxane substrate scope is an issue. However, if the paper is clearly about the properties of the catalyst that lead to enantioselectivity, then there needs to be a clearer explanation for the observed structure activity relationships.

Response:

A preliminary transition state model was shown in Fig.4 with the warning “although this is merely an illustration for one of the possible rationales.” (line 195-196).

(5) Conclusions

The conclusions are a bit strange starting from “we anticipate...”. As mentioned before, the citations given to reviews should probably be in the introduction. The citations to molecular machines reviews lack context in the conclusions (why do we want chiral molecular machines?).

Response:

Reviews on molecular devices and catalysis were shown in the introductory part and were not discussed in the conclusion.

The citations to applications in medicinal chemistry are odd because stereochemistry plays no role in any of the examples given (albeit the recognition of the sugar in 30 is stereoselective, but this is covalent stereochemistry). The authors also lump one of the few examples of an application of mechanical stereochemistry (26) in with a list of reviews when it is perhaps one of the few practical justifications (albeit preliminary) for studying how to make these molecules!

Response:

We believe that enantiopure rotaxanes with mechanical chirality is expected to contribute to create a new chemical space in medicinal chemistry, because they have never been examined in the studies directed toward medicinal chemistry. Also, because biological activity should be dependent on the enantiomeric structure of the drug candidates, supply of mechanically planar chiral rotaxanes with high enantiomeric purity should contribute to develop a new area in medicinal chemistry.

6) Some general points about terminology: “inherent chirality”, “supramolecular chirality” etc

The terminology in the area generally is a bit of a mess because it has grown up in a slightly ad hoc manner. I’m really not keen on “inherent chirality” as it’s a bit of a meaningless term – why is a calixarene (where the term is often used) more “inherently” chiral than an amino acid? I would also remove all mention of “supramolecular chirality” given that rotaxanes are technically not supramolecular species (covalent bonds must be broken to separate the components; thus they are not supramolecules). I have a preference for “mechanical chirality” or “co-conformational chirality” (depends on the type of stereogenic unit) to discuss the stereochemistry that is displayed by rotaxanes and catenanes as a result of the mechanical bond. On an unrelated point, I am not keen on the term “asymmetric synthesis” (I know it is widely used). A synthesis itself is not “asymmetric” (lacking in symmetry elements). It is an enantioselective synthesis.

Response:

The terms, “inherent chirality” and “supramolecular chirality” were replaced by “mechanical chirality”.

The term “asymmetric synthesis” was replaced by the term “enantioselective synthesis” in case of Fig. 1b, because it is actually enantioselective synthesis. The other cases, we keep the term “asymmetric” as a general term that includes

asymmetric induction and diastereomeric synthesis.

Thank you again for your pertinent suggestions.

Sincerely yours,

Takeo Kawabata
Institute for Chemical Research, Kyoto University
Uji, Kyoto 611-0011, Japan

September 18, 2020

The revised points are as follows.

For Reviewer 3:

(1) The discussion of Conformers X and Y (depicted in Fig 3). How speculative is this? Do the authors need to make these claims? Would it be better to supply in Supplementary Information as a "proposal"? I would emphasise that the main conclusion of the paper regarding the successful enantioselective kinetic resolution still holds without this discussion.

Response:

The discussion on conformers X and Y (lines 144-160) and original Fig 3 were removed.

(2) I suspect the X/Y discussion has been included as the authors may have used this model to tentatively assign the absolute structures of the rotaxanes in Fig 1(?) If so this needs to be clearly stated - highlighting uncertainties. If not, the authors should still explain how they have tentatively assigned structures of enantiomers of rotaxane.

Response:

The term "tentative assignment" was misleading. The expression was changed to "The absolute configurations of the acylated product and the recovered rotaxane alcohol were not determined".

(3)Line 36: To avoid confusion for the non-specialist, add the word "may" so line reads ". . .has been claimed that rotaxanes and catenanes may possess . . ."

Response:

Corrected as suggested.

(4) Paragraph starting Line 53: Again, as written this paragraph could mislead the non-specialist reader. The first sentence is correct in stating that "tremendous efforts

have been devoted to achieving their asymmetric synthesis". However, as written the second sentence states that construction of mechanically planar chiral rotaxanes and topologically chiral catenanes requires incorporation of additional chiral elements. This isn't true - additional chiral elements have only been required to date if attempting to prepare these molecules in mechanically chiral enriched form. This sentence (at least) needs to be re-worded.

Response:

We re-wrote as follows: lines 49-70

“Since the discovery of mechanically planar chiral rotaxanes and topologically chiral catenanes, their enantioselective preparation has been a challenge in organic synthesis. In most cases, the strategy largely relied on separation of enantiomers by HPLC with chiral stationary phases^{3,4,6}. Recently, Goldup and co-workers proposed an excellent practical method for the preparation of an enantiomeric pair of mechanically planar chiral rotaxanes via separation of the corresponding diastereomers followed by removal of the chiral auxiliary.”

(4) I would also advise the authors to check that the references included (particularly Refs 5-8) are in the correct places of this revised paragraph.

Response:

The original references 5-8 were removed because these literatures are not proper examples for topologically chiral catenanes and mechanically planar chiral rotaxanes as the referee suggested.

(5) Line 96 and Fig 2 - Why do the authors discuss 6 conformers (I to VI) of the rotaxane? Is there a reason for thinking there are 6?

Response:

Original Fig. 2c showing conformers I~IV and the corresponding discussions were removed.

(6) Line 127: What is compound 2? Structure is not provided in main manuscript.

Response:

Catalyst **2** was shown in Fig 1c (original Fig 1d).

(7) Fig 3: Correct spelling of "accessible"

Response:

Original Fig. 3 was removed.

(8) Line 200: "We unexpectedly found that enantioselectivity . . . depended on catalyst loading" - is this unexpected?

Response:

We were not sure whether it was expected or not. Therefore, the term 'unexpectedly' was removed.

(9) Ref 11: Missing paper title.

Response:

The title of the paper was shown.

Thank you again for your pertinent suggestions.

Sincerely yours,

Takeo Kawabata
Institute for Chemical Research, Kyoto University
Uji, Kyoto 611-0011, Japan

Reviewer #1 (Remarks to the Author):

I have no further suggestions.

Reviewer #2 (Remarks to the Author):

I would like to thank the authors for addressing the majority of the reviewers' concerns. There are still a few issues to address but these are mostly minor. I recommend the manuscript is accepted once these points have been addressed.

Major points

The only major point I would highlight is that the authors still don't really rationalize the effect of changing conditions/substrate on the outcome of the reaction. For example, why does catalyst concentration/equivalents make such a big difference? There is also a lot of discussion of co-conformational freedom but the results obtained from varying the steric bulk of the amide R group of the axle component show that reducing the co-conformational freedom of the system has no positive impacts on the S values? (R=Me, S=2.6; R=Ad, S=1.9). The reason for this is not discussed. Catalysts 7 & 8 with heteroaromatic substituents in place of the naphthalenes are used but the reasoning for this isn't discussed, nor is their lack of selectivity.

The most obvious answer for the effect of conditions is that there is an unselective background reaction – did the authors check the background reaction without an added catalyst? The role of the acetate anion also comes out of no-where at the end of the discussion. Could this also be responsible for some of the unusual concentration related behavior?

I should make clear that I am not looking for concrete answers here. But some discussion and clear statements to the effect of "we don't know" or "perhaps its this" would be useful to inform researchers looking to build on these results.

Finally, although Fig 4b wasn't quite what I was requesting in my review, I think it is just about acceptable. To be clear though, I was hoping for a simply molecular model using computational chemistry software. This would be no more "accurate" than the ChemDraw image but it would give a better idea of the three dimensional properties of the system and, perhaps more importantly, the relative size of substituents – this is really hard to grasp from a ChemDraw image. For instance, the "OAc" anion looks very small as drawn but I think it will occupy a large volume when viewed in a 3D model!

Minor points

I've highlighted specific sentences in the attached PDF with comments suggesting changes. I would ask that the authors take these points into account when revising the manuscript. Some additional points below:

- During the review process two new relevant reviews have been published on the synthesis of mechanically chiral molecules (<https://doi.org/10.1016/j.chempr.2020.07.012>) and mechanically interlocked catalysts (<https://pubs.acs.org/doi/abs/10.1021/acscatal.0c02032>) have been published. These should be cited in appropriate places.

- Lines 89 to 96 discuss the difficulty of discriminating mechanically planar chirality of rotaxanes, and a reference is made to a successful remote asymmetric acylation on “a substrate with a rigid conformation”; the text before is discussing rotaxanes, and the sentence after also discusses rotaxanes (Leigh chiral DMAP), but the rigid substrate is not a rotaxane. This isn’t mentioned anywhere and unless you look up the reference, reads (at least to me) like the substrate is a rotaxane.
- Figure 2d – misspelling of element as “elementt”
- Line 136 – misspelling of rotaxanes as “rotaxenes”

Reviewer #3 (Remarks to the Author):

From memory, I felt (as did the other reviewers) that the original version of this revised manuscript was of a standard suitable for publication in Nature Communications subject to revisions being undertaken. I still feel that once points have been attended the scientific results in this manuscript will be of high interest to supramolecular chemists and fully deserve to be published in Nature Communications.

Hence I will comment on the authors’ responses to the reviewers’ points of concern:

Reviewer 1:

- (1) Authors have tidied up their terminology as requested.
- (2) Authors have chosen to revise the text rather than carry out the additional experiments. For me OK - for others though . . .?
- (3) Authors have condensed their previous work as requested.
- (4) I would advise authors to detail clearly in a footnote what binding experiments were attempted and why the spectra were too complicated to analyse.
- (5) Done – minor error corrected.
- (6) Done – references added as suggested.
- (7) Done - authors have chosen to state “not determined”.
- (8) OK – but does the Abstract need changing? (Still reads up to 16 – should this be 14.6?)
- (9) OK – reasonable justification from authors I feel.

Reviewer 2:

- (1) Specific line has been revised by authors.
- (1, sic) References have been attended to as requested.
- (3) Start of R+D overlong: has been reduced by authors as requested.
- (3, sic) Start of R+D conformational effects & (4) Authors have revised these sections.
- (4, sic) Origin of selectivity: A model proposed as requested.
- (5) Conclusions: Revised in response to reviewer’s comments.
- (6) Terminology has been checked and rectified when required.

Reviewer 3:

- (1) & (2) Authors have removed the speculation regarding conformers X and Y – agree with this decision.

(3) Authors have completed minor adjustment requested.

(4) OK – introduction has been reworded.

(5) OK – references with issues have been removed.

(6) OK

(7), (8), (9) These points have been rectified.

Overall, I feel the authors have attempted to respond to all the points. As Reviewer 3 I am happy with all the responses to my previous comments.

I anticipate there may be some comeback regarding the following points from the other reviewers:

Reviewer 1 Point 2, Point 4, Point 8?, Point 9?

Reviewer 2 Point 3 (conformational effects)?

The revised points are as follows.

For reviewer 2:

Comment (1)

The only major point I would highlight is that the authors still don't really rationalize the effect of changing conditions/substrate on the outcome of the reaction. For example, why does catalyst **concentration/equivalents** make such a big difference? There is also a lot of discussion of co-conformational freedom but the results obtained from varying the steric bulk of the amide R group of the axle component show that reducing the co-conformational freedom of the system has no positive impacts on the S values? (R=Me, S=2.6; R=Ad, S=1.9). **The reason for this is not discussed. Catalysts 7 & 8 with heteroaromatic substituents** in place of the naphthalenes are used but the reasoning for this isn't discussed, nor is their lack of selectivity.

The most obvious answer for the effect of conditions is that there is an **unselective background reaction** – **did the authors check the background reaction without an added catalyst? The role of the acetate anion** also comes out of no-where at the end of the discussion. Could this also be responsible for some of the **unusual concentration related behavior**?

I should make clear that I am not looking for concrete answers here. But some discussion and clear statements to the effect of “we don't know” or “perhaps its this” would be useful to inform researchers looking to build on these results.

Response:

(1-1) On concentration effects:

We often observed that lower concentration like 0.01 M gave better results in the catalytic site-selective acylation reactions. The rationale for the concentration effects were shown in the text as shown below.

lines 156-162: “It has been often observed that lower substrate conditions like 0.01 M gave better selectivity in chemoselective acylation of diols²⁴ and geometry-selective acylation of tetrasubstituted α,α -alkenediols²⁵. We assume that substrate monomers existing in a larger ratio at the lower concentration might be more

favorable for (multi-)hydrogen bonding interaction with a catalyst for the molecular recognition process (For a possible interaction, see Fig. 4b) than substrate dimmers (hydrogen-bonded aggregate) existing in a larger ratio at the higher concentration.”

(1-2) On the dependency of catalyst equivalents:

This is an unusual phenomenon. The following comments for this issue were described in the text.

lines 193-199: “The *s* factors of the kinetic resolution of **1a** with catalyst **2** was found to depend on the catalyst loading (Fig. 4a, Table 2). Generally, the *s* factor of kinetic resolution is constant and independent from catalyst loading, when the contribution of the uncatalyzed process is negligible²⁰. Due to the lack of an uncatalyzed process under the present conditions for the kinetic resolution (Table 2, entry 1), the observation constitutes an unusual phenomenon. This could be resulting from mechanistic complexity, although the mechanistic details are totally unclear.”

(1-3) On the conformational effects for the selectivity:

We believe that catalyst **2** favors a particular conformation of rotaxane **1a** as a reactive conformer such as conformer **Y** shown in our original manuscript, irrespective its conformational stability. We tried to discuss about this issue by the discussion through conformers **X** and **Y** in our original manuscript. However, our explanation was too redundant and poor from scientific view points as referees noticed. Therefore, we removed these discussions from the manuscript.

We do not want to discuss about this issue again in the text.

(1-4) On catalysts **7** & **8** with heteroaromatic substituents:

We reported that catalysts **7** and **8** were effective for acylative asymmetric desymmetrization of *meso*-diols (reference 29). Because of the similarity of the reaction type, we examined catalysts **7** and **8** for the acylative kinetic resolution of rotaxane **1a**. Reference 29 was shown on the shoulder of **7** and **8** without explanation (line 130).

(1-5) On unselective background reaction:

There was no background reaction observed in the absence of catalyst: see Fig. 3, Table 2, entry 1.

(1-6) The role of the acetate anion and unusual concentration related behavior:

The observed concentration effects are an usual phenomenon in our site-selective organocatalytic processes as described in (1-1). The role of acetate anion is critically important in terms of both selectivity and acceleration of acylation reaction. These comments were described in text as shown below with citation of reference, 26, 27, 30, and 31.

lines 145-148: “We employed acid anhydrides as acyl donors throughout the screening because carboxylate anion generated from the anhydride and the nucleophilic catalyst was expected to be critically important for both selectivity and acceleration of the site-selective acylation^{26,27,31,32}.”

Comment (2)

Finally, although Fig 4b wasn't quite what I was requesting in my review, I think it is just about acceptable. To be clear though, I was hoping for a simply molecular model using computational chemistry software. This would be no more "accurate" than the ChemDraw image but it would give a better idea of the three dimensional properties of the system and, perhaps more importantly, the relative size of substituents – this is really hard to grasp from a ChemDraw image. For instance, the “OAc” anion looks very small as drawn but I think it will occupy a large volume when viewed in a 3D model!

Response:

A transition state model **A** was generated by an ONIOM method (M06-2X/6-31+G**:**PM6**; see SI for the detail) and shown in Fig 4b with the explanation shown below.

lines 218-226: “we propose **A** as a possible transition state model for remote asymmetric acylation (Fig. 4b). Structure **A** was generated by an ONIOM method (M06-2X/6-31+G**:**PM6**; For details, see S12) with constraint of hydrogen bonds indicated by yellow circles (Fig. 4b, right illustration) on the assumption that hydrogen bonding interaction between the amide carbonyl group of catalyst **2** and NsN-H²⁴⁻²⁶ as well as general base catalysis by the carboxylate anion located in proximity to the reacting hydroxy group³² are the keys for the effective remote asymmetric acylation. Note that transition state model **A** is generated based on the above assumption, and it does not mean the most stable transition state structure for the acylation of **1a**.”

Comments (3)

Minor points

For the suggestion indicated by the PDF attachment.

(3-1) Page 2, line 28: selectivity factor un up to 16:

This needs correcting in light of the error highlighted by the referees

Response:

Referee 1 suggested to correct the *s* value for Table 2, entry 6. “It should be 14.6, not >16!” This suggestion is correct for the reaction providing the recovered alcohol with >99.9 % ee at 71% conversion. We have corrected the *s* value of entry 6 of Table 2 to be >14.5 in place of 16.

We also observed kinetic resolution that proceeded with the *s* value of 16.1 in entry 5 of Table 2. The kinetic resolution that provides the recovered material in 63 % ee and the product in 79 % ee corresponds to the *s*-value of 16.1. The *s*-value is correct.

(3-2) Page 2, lines 34-35: It has been claimed that rotaxanes and catenanes may possess mechanical chirality^{1,2}.

not a claim - it is a scientific fact

Response:

Reference 1 and 2 discussed about proposed chiral structure of rotaxanes and catenanes without the experimental proof. We changed the expression as follows.

lines 34-35:” It has been proposed that rotaxanes and catenanes may possess mechanical chirality^{1,2}.”

The opening sentences of **Abstract** was also changed to as follows:

Asymmetric synthesis of mechanically planar chiral rotaxanes and topologically chiral catenanes has been a long-standing challenge in organic synthesis.

(3-3) Page 2, line 36: axial component: change to “axle”

Response:

We would like to use a term ‘axial component’ instead of ‘axle’ throughout the manuscript.

(3-4) Page 2, lines 38-39:

This sentence suggests a single publication by all of these authors when in fact it is two different papers.

Response:

Changed as follows:

lines 38-41: “This was eventually demonstrated by Okamoto, Sauvage, and co-workers³, and also by Okamoto, Vögtle, and co-workers⁴ based on the CD spectra of both enantiomers of topologically chiral catenanes and mechanically planar chiral rotaxanes obtained by HPLC separation of the racemates with chiral stationary phases.”

(3-5) Fig. 1 legend:

the compounds shown are not really pseudo enantiomers - the acetate group is a bit big for this to be a reasonable description. Product and recovered starting materials?

Response:

We would like to keep the term ‘pseudoenantiomeric’ because the acetate group (MW:43) seems to be small enough compared to the whole rotaxane molecule (MW of **1a**: 1179). We feel that **1a** (MW:1179) could be referred as a pseudoenantiomeric relationship with **1a-OAc** (MW:1121), because the rotaxane obtained by removal of the Ac group of **1a-OAc** in Fig 1c is completely enantiomeric to recovered **1a**.

(3-6) Page 3, line 53: discovery

first synthesis? I'm not sure they were "discovered"

Response:

References 1 and 2 proposed mechanically planar chirality and topologically chirality in the structure of rotaxanes and catenanes without experimental proof. The expression was changed as follows with newly cited reference 14:

lines 54-56: ‘Since the proposal of mechanically planar chirality and topologically chirality in the structure of rotaxanes and catenanes, respectively^{1,2}, their enantioselective preparation has been a challenge in organic synthesis¹⁴’

(3-7) Page 4, lines 67-69: there have been no reports of highly enantioselective construction of mechanically planar chiral rotaxanes or topologically chiral catenanes. I would quibble with this statement. Goldup's auxiliary strategy can deliver up to 98 : 2 diastereoselectivity, which necessarily means that one enantiomer is produced in high selectivity. I think the authors main claim here is that "there have been no reports of catalytic enantioselective synthesis in high ee"

Response:

Changed as follows according to the suggestion.

lines 67-70: ‘To date, however, there have been no reports on highly enantioselective synthesis by chiral catalysts. Here, we report the first example of highly enantioselective preparation of mechanically planar chiral rotaxanes by kinetic resolution of the racemates promoted by acylation catalysts’

(3-8) Page 7, line 120-122: Not sure how this helps? I would just remove this sentence.

Response: Our catalysts were initially designed according to the hypothesis of remote asymmetric induction. A reference for our original catalyst design was newly cited as reference 28. The sentence was changed as follows.

lines 120-124: ‘Catalyst **3** possesses uncommon structural characteristics as chiral catalysts, where chiral elements are located at a certain distance from the catalytically active site (pyridine nitrogen)²⁸. This could be somewhat suitable for remote asymmetric acylation as shown in a possible transition state model (*vide infra*: Fib 4b). ’

(3-9) Page 8, line 157: $s = 16.1$

make sure this is correct!

Response: The kinetic resolution that provides the recovered material in 63 % ee and the product in 79 % ee corresponds to the s -value of 16.1.

(3-9) Page 10, lines 190-193: additional rotatable bond seems a more likely culprit?

Response: Thank you for the suggestion. The following explanation is added.

Lines 215-216: ‘The enhanced conformational flexibility by one-carbon elongation could also be responsible for the diminished selectivity. ’

(3-10) Page 12, lines 213-215:

I'm not convinced by this rationale - the stereochemistry of these molecules is not relevant to their behaviour. I would suggest that this sentence is change to: "Rotaxanes have been shown to have biologically desirable properties (the citations given here are fine for this

statement)"The rest of the paragraph then makes sense without the slightly tenuous opening line.

Response: The expression was changed as suggested.

lines 247-248: "Rotaxanes have been shown to have biologically desirable properties³⁵⁻³⁷."

(3-11) I'd also suggest highlighting the potential applications of mechanical chirality in catalysis in the conclusions, particularly given the authors' expertise in the area of new catalyst development! There are quite a few relevant papers here (some of which are already cited elsewhere). Obvious authors to look up are Niemeyer, Takata (some lovely work here!), Leigh, Berna and Goldup.

Response: The following sentence was added.

lines 245-246: "It would also contribute to potential applications of mechanical chirality in catalysis⁶⁻¹³."

Comment (4)

During the review process two new relevant reviews have been published on the synthesis of mechanically chiral molecules (<https://doi.org/10.1016/j.chempr.2020.07.012>) and mechanically interlocked catalysts (<https://pubs.acs.org/doi/abs/10.1021/acscatal.0c02032>) have been published. These should be cited in appropriate places.

Response:

Second one (ACS Catalysis, 9, 7719-7733 (2020) was already cited in reference 9.

The first one was cited in reference 14.

Comment (5)

Lines 89 to 96 discuss the difficulty of discriminating mechanically planar chirality of rotaxanes, and a reference is made to a successful remote asymmetric acylation on "a substrate with a rigid conformation"; the text before is discussing rotaxanes, and the sentence after also discusses rotaxanes (Leigh chiral DMAP), but the rigid substrate is not a rotaxane. This isn't mentioned anywhere and unless you look up the reference, reads (at least to me) like the substrate is a rotaxane.

Response: The expression was changed as follows.

lines 94-96: "a highly successful example for remote asymmetric acylation of a bisphenol compound with rigid conformation has been reported²²"

Comment (6)

Figure 2d – misspelling of element as "element"

- Line 136 – misspelling of rotaxanes as "rotaxenes"

Response:

Corrected as suggested.

For reviewer 3:

(4) I would advise authors to detail clearly in a footnote what binding experiments were attempted and why the spectra were too complicated to analyze.

Response:

NOESY, REOSY, and COSY spectra of a mixture of catalyst **2** and racemic rotaxane **1a** were measured at $-40\text{ }^{\circ}\text{C}$ with a 600 MHz NMR instrument. However, the spectra were too complicated to analyze. As far as we could analyze, no positive information was obtained, which suggests interaction or binding between **2** and **1a**.

The spectra were submitted as review-only materials (file name: binding experiments.pdf). We would not like to mention about these issues in the text or SI, because no meaningful information was involved.

Thank you again for your pertinent and generous suggestions.

Sincerely yours,

Takeo Kawabata
Institute for Chemical Research, Kyoto University
Uji, Kyoto 611-0011, Japan

tel: +81-774-38-3190: fax: +81-774-38-3197
e-mail: kawabata@scl.kyoto-u.ac.jp